# Comparison of Heart Rate Feedback from Dry-Electrode ECG, 3-Lead ECG, and Pulse Oximetry during Newborn Resuscitation

**DOI:** 10.3390/children8121092

**Published:** 2021-11-26

**Authors:** Siren Rettedal, Joar Eilevstjønn, Amalie Kibsgaard, Jan Terje Kvaløy, Hege Ersdal

**Affiliations:** 1Department of Paediatrics, Stavanger University Hospital, 4011 Stavanger, Norway; amalie.kibsgaard-petersen@sus.no; 2Faculty of Health Sciences, University of Stavanger, 4021 Stavanger, Norway; hege.ersdal@safer.net; 3Laerdal Medical, 4002 Stavanger, Norway; Joar.Eilevstjonn@laerdal.com; 4Department of Research, Section of Biostatistics, Stavanger University Hospital, 4011 Stavanger, Norway; jan.t.kvaloy@uis.no; 5Department of Mathematics and Physics, University of Stavanger, 4021 Stavanger, Norway; 6Critical Care and Anaesthesiology Research Group, Stavanger University Hospital, 4011 Stavanger, Norway

**Keywords:** NeoBeat, heart rate monitoring, Pulse Oximetry, newborn resuscitation, ECG, dry-electrode technology, resuscitation guidelines, electrocardiogram, neonatal resuscitation, heart rate assessment

## Abstract

Background: Assessment of heart rate (HR) is essential during newborn resuscitation, and comparison of dry-electrode ECG technology to standard monitoring by 3-lead ECG and Pulse Oximetry (PO) is lacking. Methods: NeoBeat, ECG, and PO were applied to newborns resuscitated at birth. Resuscitations were video recorded, and HR was registered every second. Results: Device placement time from birth was median (quartiles) 6 (4, 18) seconds for NeoBeat versus 138 (97, 181) seconds for ECG and 152 (103, 216) seconds for PO. Time to first HR presentation from birth was 22 (13, 45) seconds for NeoBeat versus 171 (129, 239) seconds for ECG and 270 (185, 357) seconds for PO. Proportion of time with HR feedback from NeoBeat during resuscitation from birth was 85 (69, 93)%, from arrival at the resuscitation table 98 (85, 100)%, and during positive pressure ventilation 100 (95, 100)%. For ECG, these proportions were, 25 (0, 43)%, 28 (0, 56)%, and 33 (0, 66)% and for PO, 0 (0, 16)%, 0 (0, 16)%, and 0 (0, 18)%. All *p* < 0.0001. Conclusions: NeoBeat was faster to place, presented HR more rapidly, and provided feedback on HR for a larger proportion of time during ongoing resuscitation compared to 3-lead ECG and PO.

## 1. Introduction

After birth, the newborn heart rate (HR) is important in the assessment of the effectiveness of spontaneous breathing, need for positive pressure ventilation (PPV), and for determining the response to resuscitation. International guidelines recommend that HR should be assessed and PPV initiated within 60 s of birth if the newborn is not breathing or is gasping or if HR is <100 beats per minute (bpm) [1,2]. Therefore, a rapid and reliable method of measuring the newborn HR is critical during newborn resuscitation [1]. Use of auscultation by stethoscope is still the most common method for evaluating HR in the newborn during resuscitation, but the need for continuous HR monitoring requires advanced modalities, such as electrocardiography ECG and Pulse Oximetry (PO). ECG provides reliable feedback on HR faster than PO [3] and is suggested for monitoring of HR during newborn resuscitation [1]. New devices for ECG monitoring during newborn resuscitation have been developed and include the dry-electrode ECG technology used in NeoBeat Newborn Heart Rate Meter (Laerdal Global Health, Stavanger, Norway) (Figure 1) [1,4,5]. The novel dry-electrode technology for HR monitoring of newborns immediately after birth and continuously during resuscitation is currently mostly used in research settings, a few places around the world. Comparison of this new technology to the current gold-standard 3-lead ECG monitoring during newborn resuscitation has been identified as a knowledge gap [1]. HR obtained by NeoBeat correlated well with that of 3-lead ECG in healthy newborns [6] and was faster to place and acquired HR more rapidly compared to 3-lead ECG and PO in a small cohort of preterm and term newborns requiring minimal resuscitation [7]. However, no studies have compared the devices in real-life settings with unanticipated and complicated resuscitations performed by non-research staff. 

In this study, we evaluated a new concept to advance newborn resuscitations. Studying newborns with gestational age (GA) ≥ 32 weeks receiving resuscitation with PPV after birth, we compared the dry-electrode ECG technology NeoBeat to standard monitoring with 3-lead ECG and PO; for (i) time from birth to placement of the devices, (ii) device placement time, (iii) time to HR presentation, (iv) proportion of time during resuscitation with HR feedback, and (v) HR correlation between devices.

## 2. Materials and Methods

### 2.1. Setting

Data for this prospective descriptive study were collected from June 2019 through February 2021 at Stavanger University Hospital, Norway. It is the only hospital in the region that provides both obstetric and neonatal services for approximately 4300 annual deliveries and newborns GA ≥ 23 weeks, and it is well-suited for population-based studies. Resuscitation of newborns at birth occurs at three resuscitation stations: the resuscitation room on the labour ward, the caesarean section operating theatre, and the midwife-run delivery unit. Overall rate of PPV provision at birth is 3.6% [8]. A midwife, midwife assistant, and paediatric resident initiate resuscitations, with the addition of a neonatologist, neonatal nurse, and anaesthetic team for complicated resuscitations. National resuscitation guidelines are based on the International Liaison Committee on Resuscitation and European Resuscitation Council guidelines on newborn resuscitation [1,2]. Healthcare providers (HCPs) undergo regular skill and team simulation training.

### 2.2. Equipment and Data Collection

During the study period, NeoBeat was attached to all newborns by the midwife assistant immediately after birth for HR detection (Figure 1). NeoBeat can be placed directly on the thorax or abdomen without prior drying of the skin and uses dry-electrode ECG technology to pick up and calculate the HR. The HR was displayed in clear view to the HCPs and was also sent to the Liveborn mobile device application (Laerdal Global Health, Stavanger, Norway) using Bluetooth. Since NeoBeat is intended for newborns of approximate size ≥ 1500 g, we included newborns with GA ≥ 32 weeks. Newborns delivered by caesarean section had NeoBeat applied when free of the sterile operating field or on the resuscitation table. The time of birth was registered in the Liveborn application by the midwife assistant. In non-breathing, gasping, or inadequately breathing newborns not responding to drying and stimulation, the umbilical cord was clamped, and the newborn immediately transferred to the resuscitation room with the wireless NeoBeat still attached. The HR was shown on the NeoBeat display. HCPs were instructed to attach standard HR monitoring with 3-lead ECG and PO (Carescape Patient monitor B450 or B650, GE Healthcare, Boston, MA, USA) to all newborns receiving PPV, when time permitted after the newborn arrived at the resuscitation table. The sequential order of sensor application was left to the discretion of the attending HCPs. For most resuscitations, three or more HCPs were present from the start to assist in simultaneous placement of devices. HR from ECG and PO was displayed on the monitor screen and video recorded for subsequent HR extraction. 

Resuscitations were video recorded using cameras placed above the resuscitation tables, capturing the newborn and the hands of the HCPs. The clock in the video server (ECG and PO HR) and timestamp in the Liveborn application (NeoBeat HR) were automatically synchronized daily. Each patient dataset was checked manually for signal integrity by two investigators (S.R., J.E.). Patient characteristics were automatically extracted from the electronic medical records. 

### 2.3. Inclusion and Exclusion Criteria

Inclusion criteria were newborns with GA ≥ 32 weeks that received PPV within the first five minutes after birth. Only newborns with complete datasets with video recordings of the resuscitation table, HR signal data from NeoBeat, 3-lead ECG and PO, and exact time of birth registration in the Liveborn application were included. Newborns with congenital malformations that interfered with the placement of device sensors were excluded. 

### 2.4. Calculations and Definitions

Video recordings were reviewed by two investigators (S.R., A.K.) using XProtect Smart Client software version 2016 (Milestone, Copenhagen, Denmark). The time of arrival at the resuscitation table, placement time of NeoBeat (when not applied in the delivery room), 3-lead ECG and PO, time of initiation and discontinuation of PPV, intubation, chest compressions, and number of HCPs present when the newborn was placed on the resuscitation table were registered. Time to initiation of PPV was defined as time from birth to the first positive pressure inflation. Duration of PPV was defined as time from the first to last positive pressure inflation or, if intubated, when the newborn was moved from the resuscitation table to the transport incubator. ECG device placement time was calculated from wiping of the chest or picking up the sensor to affixing all three gel-electrodes and PO device-placement time from initiation of drying the right wrist or picking up the sensor until affixing it or plugging it into the monitor, whichever came last. NeoBeat placement time was calculated from the time of picking up the sensor until applying it around the newborns’ thorax or abdomen. Time from birth to placement of NeoBeat was calculated as the first time of contact between the dry-electrodes and skin as registered by NeoBeat. HR was registered every second throughout the resuscitation for all devices. Proportion of resuscitation with HR feedback was defined as percentage of time with HR feedback from the time of birth, arrival at the resuscitation table, and from initiation of PPV to discontinuation of PPV. First heart rate was calculated as the median of first eleven HRs detected from NeoBeat. 

### 2.5. Statistical Analysis

Continuous data were summarized by median and quartiles or minimum and maximum, unless otherwise stated. Differences in time or differences in proportion of time between the devices were tested by the Wilcoxon paired samples test. To illustrate agreement between the devices in heart rate monitoring, box-plots of differences over time were constructed. A two-sided *p* < 0.05 defined statistical significance. 

Data processing, data point extraction, and statistical analysis were done using MATLAB R2021a (MathWorks, Natick, MA, USA) and R version 4.1.0 (R Development Core Team, Vienna, Austria). 

### 2.6. Ethical Considerations

This descriptive study was part of the research consortium Safer Births Bundle Stavanger on newborn resuscitation, including a randomized controlled trial (NCT03849781), with ethical approval from the regional ethical committee (reference number 2018/338). Expectant mothers were invited to participate in the study at routine ultrasound screening in pregnancy week 20 or when admitted for labour. HCPs could decline participation, and the video recordings in these instances were not included in the analysis. 

## 3. Results

### 3.1. General Characteristics of Participants

Of 7268 newborns born during the study period with GA ≥ 32 weeks, 248 (3.4%) were resuscitated with PPV, of which 206 had consent for participation. Of these, 48 newborns had complete datasets and were included in the study. The time from birth to arrival at the resuscitation table was 44 (32, 74) seconds, to initiation of PPV 70 (46, 121) seconds, and duration of PPV from first to last inflation 153 (74, 257) seconds. A minority required chest compressions (*n* = 4) and/or intubation (*n* = 2). NeoBeat was kept in place during chest compressions and displayed HR throughout. The first HR, as detected by NeoBeat, ranged from minimum 45 to maximum 193 bpm, with median 109 (78, 144) bpm at median 22 (13, 45) seconds after birth. A total of 56% of resuscitated newborns had first HR > 100 bpm. There were no adverse effects from the placement of NeoBeat, 3-lead ECG, or PO in any of the 48 newborns. In 45 resuscitations, three or more HCPs were present when the newborn arrived at the resuscitation table; in the remaining three cases, two HCPs initiated resuscitation. The characteristics of the newborns are shown in Table 1. 

### 3.2. Time from Birth to Placement of the Devices and Device Placement Time

Time from birth to device placement was median 6 (4, 18) seconds for NeoBeat versus 138 (97, 181) seconds for ECG and 152 (103, 216) seconds for PO (all *p* < 0.0001). In nine cases, mostly after caesarean sections, NeoBeat was placed on the newborn at the resuscitation table with video capturing, and device placement time was 2 (2, 3) seconds versus 25 (18, 31) seconds for ECG (*n* = 48) and 24 (18, 32) seconds for PO (*n* = 45), as shown in Figure 2a. 

### 3.3. Time to First HR Presentation 

Time in seconds from device placement to first HR presentation was 10 (6, 20) for NeoBeat versus 20 (14, 69) for ECG (*p* = 0.0001) and 96 (28, 193) for PO (*p* < 0.0001) (Figure 2b). Time in seconds from birth to first HR presentation was 22 (13, 45) for NeoBeat versus 171 (129, 239) for ECG and 270 (185, 357) for PO (Figure 3a). All *p* < 0.0001. Time in seconds to first HR presentation from the newborn arrived at the resuscitation table is presented in Figure 3b, all *p* < 0.0001. In total, 38 of the 48 resuscitated newborns had HR feedback prior to initiation of PPV. In the 10 cases where PPV was initiated before HR was presented, seven had first HR < 100 bpm.

### 3.4. Proportion of Time with HR Feedback during Resuscitation

Median proportion of time with HR feedback from NeoBeat from the time of birth until discontinuation of PPV was 85 (69, 93)% versus 25 (0, 43)% for ECG and 0 (0, 16)% for PO (Figure 4a). Proportion of time with HR feedback from NeoBeat during resuscitation, from the time of arrival at the resuscitation table, was 98 (85, 100)% versus 28 (0, 56)% for ECG and 0 (0, 16)% for PO (Figure 4b). Lastly, proportion of time with HR feedback from NeoBeat during PPV was 100 (95, 100)% versus 33 (0, 66)% for ECG and 0 (0, 18)% for PO (Figure 4c). All *p* < 0.0001. 

A timeline illustrating each newborn and resuscitative actions, including placement on the resuscitation table, duration of PPV and presentation of first HR as function of time since birth is presented in Figure 5.

### 3.5. Heart Rate Correlation between Devices

Box plots showing the difference in HR over time between the three devices are shown in Figure 6. HR obtained by NeoBeat versus ECG shows a difference of 1 (−2, 3) bpm (*n* = 3940) and the number of data pairs with difference > 10 bpm was 383 (9.7%). PO tended to underestimate HR until six minutes after birth. HR obtained by PO versus ECG showed a difference of −1 (−57, 1) bpm (*n* = 1214); the number of data pairs with difference > 10 bpm was 393 (32.4%). HR obtained by PO versus NeoBeat showed a difference of −2 (−10, 1) bpm (*n* = 2286); the number of data pairs with difference > 10 bpm was 585 (25.6%). 

## 4. Discussion

This is the first study comparing HR detection properties of dry-electrode ECG technology to standard monitoring with 3-lead ECG and PO during real-life resuscitations performed by non-research personnel in moderately to severely compromised newborns. The rationale for conducting the study was threefold: (1) early and continuous feedback on HR is considered the most sensitive indicator of the newborns’ condition during resuscitation; (2) current standard HR monitoring technology poses feasibility and latency challenges; (and 3) HR detection properties of newly developed dry-electrode ECG technology should be compared to current gold-standard 3-lead ECG.

The dry-electrode device NeoBeat was faster to place, presented feedback on HR more rapidly, and presented HR for a larger proportion of time during resuscitation compared to 3-lead ECG and PO. HR, as detected by PO, was significantly lower for the first six minutes after birth compared to HR as detected by NeoBeat and 3-lead ECG. First measured HR by NeoBeat at a median of 22 s after birth exceeded 100 bpm in 56% of resuscitated newborns.

### 4.1. NeoBeat Was Placed More Rapidly

Most studies comparing two or more HR assessment technologies, for time to devices placement and time to first HR presentation, studied preterm infants or low-risk newborns not requiring resuscitation at birth [9,10]. Only one previous study compared time to devices placement and time to first HR presentation between dry-electrode ECG and standard monitoring with 3-lead ECG and PO during newborn stabilization and resuscitation [7]. In a heterogeneous cohort of 28 newborns of GA 34.6 (30.1, 40.9) weeks, where nine received PPV, time to device placement after arrival at the resuscitation table was significantly shorter with NeoBeat, only three seconds compared to 16 s for ECG and 20 s for PO. Furthermore, total time from initiation of device placement to HR presentation was significantly shorter with NeoBeat [7]. This is in line with the observations in our cohort of mainly term newborns, all in need of PPV.

### 4.2. Dry-Electrode Technology Provided HR Feedback More Rapidly and More Continuously during Resuscitation

The most important finding in this study was that dry-electrode ECG consistently presented an accurate and reliable HR more rapidly and more continuously than the other devices. This was the case during the critical time immediately after birth, through transfer to the resuscitation table, and during resuscitations with PPV, intubation, and/or chest compressions. An advantage of the dry-electrode technology is that the device can be placed on the newborn immediately after birth in the delivery room and provides an accurate HR within seconds, facilitating decision making, earlier intervention, and later, corrective actions during PPV as necessary (Figure 5). Furthermore, the wireless design allows for continuous monitoring during transfer to the resuscitation table and during periods of chest compressions. Continuous HR assessment can alternatively be obtained by dedicating one member of the team to apply 3-lead ECG, PO, or auscultate by stethoscope, but this places a higher demand on required number of members in a resuscitation team and may delay initiation of PPV. NeoBeat is a reusable and consumable-free low-cost device, easy to use by a single care provider. This is important, as the vast majority of neonatal deaths related to birth asphyxia occur in low- and middle-resource settings, where costly ECG and PO equipment are mostly unavailable, and staffing levels are low.

### 4.3. Current Guideline Recommendations and HR Thresholds

Newborn resuscitation guidelines recommend initiation of PPV if HR is <100 bpm and start of chest compression if HR is <60 bpm. These predefined HR thresholds are not supported by rigorous scientific evidence but rather based on expert consensus [1,10]. Updated normograms based on dry-electrode ECG technology show that in healthy, spontaneously breathing newborns, initial HR increases rapidly from 120 to 170 bpm during the first 5 to 30 s after birth, peaking at 175 bpm at one minute, before slowly decreasing to 165 five minutes after birth [4,5]. It is surprising that in this cohort of compromised newborns, where all received PPV, and 10% received chest compressions and/or intubation, 56% had a first detected HR exceeding 100 bpm at a median 22 s after birth. A similar HR distribution has been described in a large observational study of 1237 resuscitated newborns in Tanzania using a predecessor of NeoBeat. In this study, 51% of the apnoeic newborns had an initial HR > 100 bpm, measured after stimulation at a median 102 s after birth and prior to initiation of PPV [11]. These findings from both Tanzania and our high-resource setting indicate that half of depressed newborns in need of urgent PPV have an initial HR > 100 bpm. Another study from Tanzania, in a setting without an intensive care unit, demonstrated how the risk of death or prolonged admission increases by 16% for every 30 s delay in initiating PPV up to six minutes [12]. Furthermore, PPV with adequate tidal volumes results in a rapid increase in HR in apnoeic newborns born at term [13]. These findings suggest that initial HR assessment and considerations related to the arbitrary threshold of 100 bpm should not delay initiation of PPV in non-breathing newborns.

### 4.4. PO Underestimates HR until Six Minutes from Birth

PO measures pulse waves and underestimated HR until six minutes from birth compared to NeoBeat and ECG in our study, probably due to poor peripheral perfusion in compromised newborns and low left ventricular outflow during transition. Our findings are supported by previous studies, where HR detected by PO was significantly lower than HR detected by ECG until seven minutes after birth [3]. If HR is presented late or inaccurately, it can delay critical interventions such as the need for increased ventilation volume or other corrective measures due to ineffective PPV or lead to inappropriate interventions such as chest compression.

### 4.5. Prediction Value of HR for Outcome

Initial HR and HR response to resuscitation can provide important predictive information on newborns at greatest risk. Previous studies have shown that an initial rapid increase in HR to >100 bpm in response to PPV is associated with 75% reduction in risk of death, whereas a decrease in HR to <100 bpm during pauses in PPV is associated with an almost two-fold increased risk of death in a low-resource setting [14]. In most hospitals in high-resource settings, the majority of newborns receiving resuscitation with PPV are returned to parental care, and only a minority are admitted to the neonatal intensive care unit [8]. Objective information on initial HR and HR response to resuscitation may help identify newborns at high risk who should be triaged to closer follow-up, monitoring, or neonatal intensive care admission after resuscitation.

### 4.6. Future Studies and Knowledge Gaps

Dry-electrode ECG technology was superior for HR detection at birth and during resuscitation in this cohort of moderately to severely compromised newborns. However, there is still a need to investigate potential clinical benefits of rapid and continuous HR feedback on, e.g., time to initiation of resuscitative interventions and short term outcomes. Our research group has undertaken a larger randomized controlled trial to address these knowledge gaps.

### 4.7. Strengths and Limitations

The strengths of this study include the design with real-life resuscitations of mainly term newborns, performed by non-research staff. Limitations of this study were loss of participants, as only newborns with complete datasets (video recording of the resuscitation, HR signal data from dry-electrode technology, 3-lead ECG and PO, and exact time of birth registration in the Liveborn application) were included. This may have resulted in a selection of the most complex resuscitations, where time permitted application of all three sensors. Feedback on HR from NeoBeat may have reduced the motivation to apply ECG and PO although this was standard procedure during the study. Furthermore, serial rather than simultaneous application of ECG and PO may have contributed to the results although in the majority of cases, ECG and PO sensors were applied by two individual HCPs. Registration of exact time of birth in the Liveborn application is prone to human error, and any drift in timestamp between the video server and Liveborn application may have caused non-alignment of HR data between devices. This was corrected for manually by checking and, if needed, annotating HR from NeoBeat in the video to ensure alignment.

## 5. Conclusions

NeoBeat was faster to place, presented HR more rapidly, and provided feedback on HR for a larger proportion of time during ongoing resuscitation compared to 3-lead ECG and PO. HR correlated well with that obtained from the gold-standard 3-lead ECG. Future research should focus on assessing the impact of the routine use of HR feedback from birth and during newborn resuscitation on timing and quality of PPV and short-term outcomes.

## Figures and Tables

**Figure 1 children-08-01092-f001:**
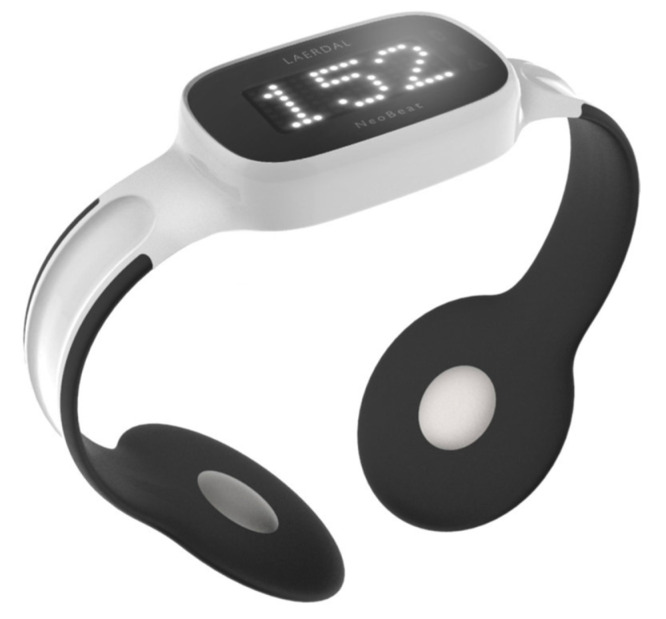
NeoBeat Newborn Heart Rate Meter (Laerdal Global Health, Stavanger, Norway) low-cost, dry-electrode ECG technology, produced for heart rate monitoring of newborns immediately after birth and continuously during resuscitation.

**Figure 2 children-08-01092-f002:**
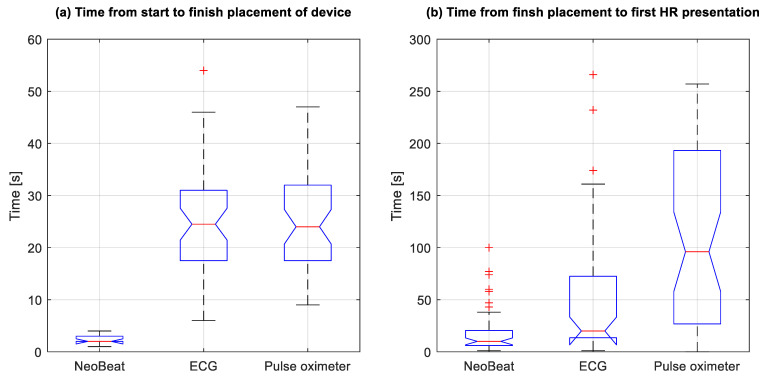
Box plots with comparison of time (**a**) from start to finish placement and (**b**) from finish placement to first HR presentation. Four of 48 cases did not apply Pulse Oximetry before after PPV was discontinued. HR, heart rate. ECG, electrocardiogram. + represents outliers. Note for (**b**): One pulse oximeter outlier at 768 s is not shown.

**Figure 3 children-08-01092-f003:**
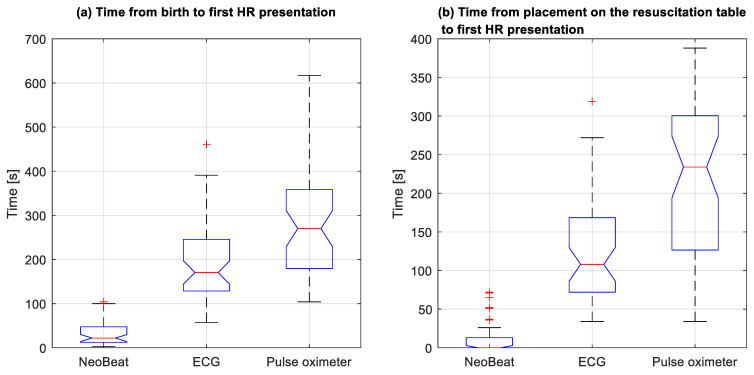
Box plots with comparison of time to first heart rate presentation from (**a**) the time of birth and (**b**) placement on the resuscitation table. HR, heart rate. Note: One pulse oximeter outlier at 1005 s in (**a**) and 963 s in (**b**) is not shown.

**Figure 4 children-08-01092-f004:**
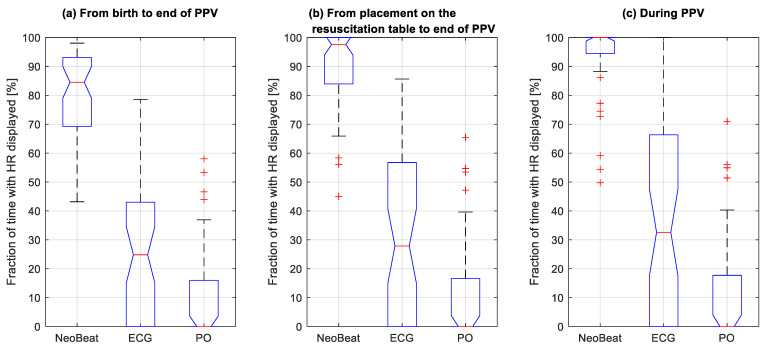
Box plots comparing fraction of time in percentage with heart rate displayed for the three devices among resuscitated newborns, from (**a**) the time of birth, (**b**) placement on the resuscitation table, and (**c**) during positive pressure ventilation. HR, heart rate; PO, pulse oximeter; PPV, positive pressure ventilation. Dots represents outliers.

**Figure 5 children-08-01092-f005:**
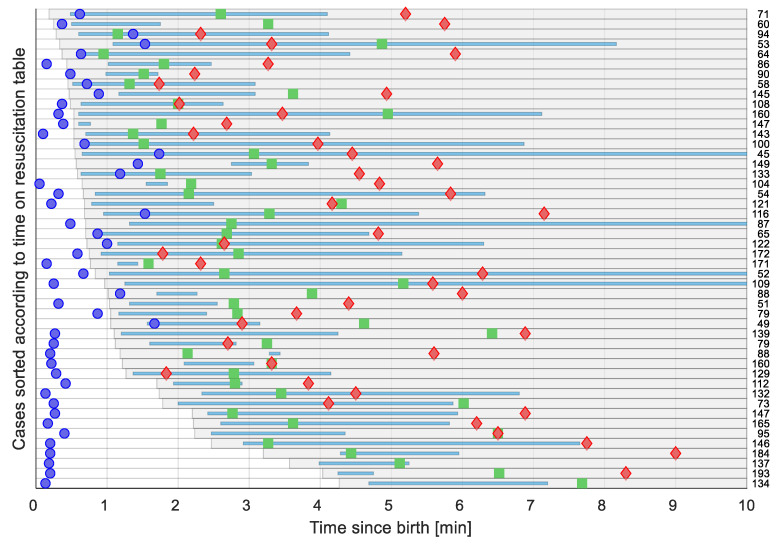
Timelines illustrating the 48 resuscitated newborns sorted according to time of placement on the resuscitation table from the time of birth. Each grey row represents time on the resuscitation table. Blue lines represent duration of PPV. First acquired heart rate is presented as blue dots for NeoBeat, green squares for gel-electrode ECG, and red diamonds for Pulse Oximetry. Number on the right-hand edge is the first heart rate in bpm for each case.

**Figure 6 children-08-01092-f006:**
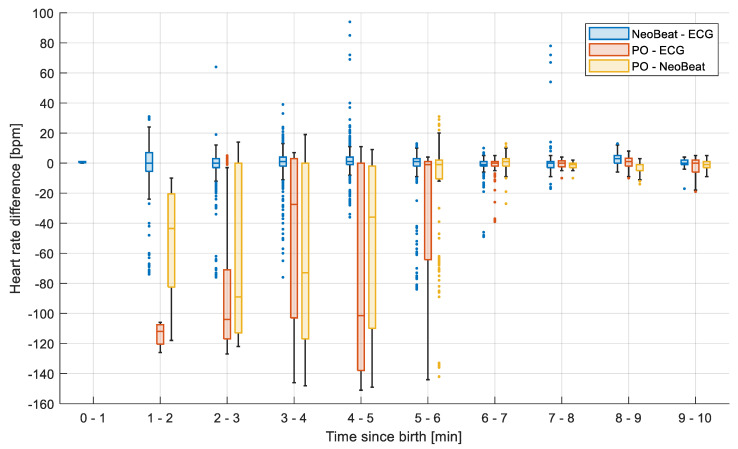
Box plots showing the difference in heart rate over time between the various devices (NeoBeat–ECG, *n* = 3953; PO–ECG, *n* = 1214; PO–NeoBeat, *n* = 2290). PO tends to underestimate HR the initial five-six minutes during resuscitation, whereas NeoBeat and gel-electrode ECG correlate well. PO, Pulse Oximetry. Dots represents outliers.

**Table 1 children-08-01092-t001:** Patient demographics of 48 newborns receiving PPV.

Patient Demographic	
Gestational age (weeks)	
Mean (SD)	39.5 (1.8)
Median (range)	40 (32–41)
Term	45
Late preterm (34 0/7–36 6/7 weeks)	2
Preterm < 34 weeks	1
Birth weight (grams)	
Mean (SD)	3538 (609)
Median (range)	3583 (1730–5015)
Female sex	20 (40%)
Mode of delivery	
Spontaneous vaginal	13 (26%)
Instrumental vaginal	26 (52%)
Forceps	4 (8%)
Vacuum	22 (44%)
Emergency Caesarean section	11 (22%)
Apgar score	
Median (range)	
1 min	4 (0–9)
5 min	7 (1–10)
10 min	9 (4–10)
Umbilical arterial pH	
Mean (SD)	7.2 (0.09)
Intubation	2 (4%)
Chest compressions	4 (8%)

Gender, mode of delivery, intubation, and chest compressions are given as *n* (%). PPV, positive pressure ventilation.

## Data Availability

The data presented in this study are available on request from the corresponding author. The data are not publicly available due to privacy statements made in informed consent obtained from participants.

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
