# Peer review of "Comparison of Heart Rate Feedback from Dry-Electrode ECG, 3-Lead ECG, and Pulse Oximetry during Newborn Resuscitation"

_children, 2021, doi:10.3390/children8121092_

Round 1
Reviewer 1 Report
In this prospective descriptive study authors compared the dry-electrode ECG technology NeoBeat to standard monitoring with 3-lead ECG and pulse-oxymetry, in newborns with gestational age ≥32 weeks receiving resuscitation after birth. NeoBeat was faster to place, presented HR more rapidly, and provided feedback on HR for a larger proportion of time during ongoing resuscitation, compared to 3-lead ECG and PO. Thus, NeoBeat could be a promising tool to accurately detect HR in the first minutes of life and to appropriately guide newborn's resuscitation steps.
I have only some comments:
1) NeoBeat is a new device so it could be useful for readers to have some more description in the Materials and Methods section as: how was it positioned precisely (abdomen or thorax)? the minimum birth weight and gestational age in study population were 1730g and 32 weeks gestation respectively, is it suitable for smaller newborns (or it is too big for newborns less than 30w GA)? babies under 32 weeks gestation should be kept inside a polyethilene bag without drying, could it be difficult to apply on a wet and slippery surface? In nine of 48 cases NeoBeat was placed on the newborn at the resuscitation table and not immediately after birth, why?
2) authors state that HR from ECG and PO was displayed on the monitor screen (Carescape Patient monitor) and video recorded, using motion sensor cameras placed above the resuscitation tables, for subsequent HR extraction. How was the HR video recorded from Carescape monitor precisely synchronized with the HR from NeoBeat?
3) line 156 "reminding" should be "remaining", please correct.
4) authors stated that "....the vast majority of neonatal deaths
related to birth asphyxia occur in low- and middle resource settings, where staffing levels are low, and costly ECG and PO equipment mostly unavailable". Do you mean that NeoBeat is cost effective in respect to standard ECG (as PO is anyway necessary for SaO2 monitoring)?
Author Response
Thank you for your valuable and constructive feedback. Changes are made according to the reviewer`s suggestions, see below.
1) Method section 2.2. Equipment and data collection line 83-90
"NeoBeat can be placed directly on the thorax or abdomen without prior drying of the skin, and uses dry-electrode ECG technology to pick up and calculate the HR. The HR was displayed in clear view to the HCPs, and was also sent to the Liveborn mobile device application (Laerdal Global Health, Stavanger, Norway) using Bluetooth. Since NeoBeat is intended for newborns of approximate size ≥1500 grams, we included newborns with GA≥32 weeks. Newborns delivered by caesarean section had NeoBeat applied when free of the sterile operating field or on the resuscitation table."
3.1.2. Time from birth to placement of the devices and device placement time line 169-173
"In nine cases, mostly after caesarean sections, NeoBeat was placed on the newborn at the resuscitation table with video capturing, and device placement time was 2 (2, 3) seconds, versus 25 (18, 31) seconds for ECG (n=48) and 24 (18, 32) seconds for PO (n=45), as shown in Figure 2a."
2) Method section 2.2. Equipment and data collection line 102-105
"Resuscitations were video recorded using cameras placed above the resuscitation tables, capturing the newborn and the hands of the HCPs. The clock in the video server (ECG and PO HR) and timestamp in the Liveborn application (NeoBeat HR) were automatically synchronized daily."
3) Correction line 164 to "remaining".
4) 4.2. Dry-electrode technology provided HR feedback more rapidly and more continuously during resuscitation line 267-270.
"NeoBeat is a reusable and consumable-free low-cost device, easy to use by a single care provider. This is important, as the vast majority of neonatal deaths related to birth asphyxia occur in low- and middle resource settings, where costly ECG and PO equipment are mostly unavailable and staffing levels are low."
Reviewer 2 Report
Well-written manuscript.
Minor comments:
Introduction:
Use of auscultation by stethoscope is not mentioned in the introduction, and the need for “continuous” HR monitoring is what requires advanced modalities such as EKG, PO or NeoBeat does not come across.
Line 161: 152 seconds appears to be a very long time to place PO from time of birth. How do the authors explain this? Could this be because the EKG was routinely placed prior to application of PO?
Line 161: “All p<0.0001” may be included at the end of prior sentence within parenthesis.
Line 206: “show” should be “shown”.
Lines 315-316: Serial rather than simultaneous application of EKG lead and PO may have contributed to the results as well and may be added as a limitation.
Author Response
Thank you for the constructive and valuable feedback. We have made changes according to your suggestions.
1) Introduction line 40-42
"Use of auscultation by stethoscope is still the most common method for evaluating HR in the newborn during resuscitation, but the need for continuous HR monitoring requires advanced modalities such as electrocardiography ECG and pulse oximetry (PO)."
2) As the reviewer points out, placement of ECG were more frequently prioritized compared to PO, but the order of placement was left to the discretion of the HCPs. Int the majority of cases two different HCPs applied ECG and PO.
2.2. Equipment and data collection line 94-101
"The HR was shown on the NeoBeat display. HCPs were instructed to attach standard HR monitoring with 3-lead ECG and PO (Carescape Patient monitor B450 or B650, GE Healthcare, Boston, MA) to all newborns receiving PPV, when time permitted after the newborn arrived at the resuscitation table. The sequential order of sensor application was left to the discretion of the attending HCPs. For most resuscitations, three or more HCPs were present from the start to assist in simultaneous placement of devices."
3) 3.1.2. Time from birth to placement of the devices and device placement time line 169
"Time from birth to device placement was median 6 (4, 18) seconds for NeoBeat versus 138 (97, 181) seconds for ECG and 152 (103, 216) seconds for PO (all p<0.0001)."
4) 3.1.5. Heart rate correlation between devices line 215
Show corrected to "shown".
5) 4.7. Strengths and Limitations line 328-331
"Furthermore, serial rather than simultaneous application of ECG and PO may have contributed to the results, although in the majority of cases ECG and PO sensors were applied by two individual HCPs."